# A cost effective machine learning based network intrusion detection system using Raspberry Pi for real time analysis

R.W.K.S. Wijethilaka©, Kanishka Yapa, Deemantha Siriwardena ©*

Faculty of Computing, Sri Lanka Institute of Information Technology, Malabe, Sri Lanka

© These authors contributed equally to this work.
* deemantha.s@sliit.lk

## Abstract

In an increasingly interconnected world, the security of sensitive data and critical operations is paramount. This study presents the development of a Network Intrusion Detection System (NIDS) that analyzes both inbound and outbound network traffic to detect and classify various cyber attacks. The research begins with an extensive review of existing intrusion detection techniques, highlighting the limitations of traditional methods when addressing the unique security challenges posed by distributed networks. To overcome these limitations, advanced machine learning algorithms, including Random Forest, Long Short Term Memory (LSTM) networks, Artificial Neural Networks (ANN), XGBoost, and Naive Bayes, are employed to create a robust and adaptive intrusion detection system. The practical implementation utilizes a Raspberry Pi as the central processing unit for real time traffic analysis, supported by hardware components such as Ethernet cables, LEDs, and buzzers for continuous monitoring and immediate threat response. A comprehensive alert system is developed, sending email notifications to administrators and activating physical indicators to signify detected threats. Our proposed NIDS achieves 96.5 detection accuracy on the NF-UQ-NIDS dataset, with a significantly reduced false positive rate after applying SMOTE. The system processes real time network traffic with an average response time of 50 milliseconds, outperforming traditional IDS solutions in accuracy and efficiency. Evaluation using the NF-UQ-NIDS dataset demonstrates a significant improvement in detection accuracy and response time, establishing the system as an effective tool for safeguarding networks against emerging cyber threats.

## Introduction

As cyber threats continue to evolve in sophistication and frequency, the imperative for robust security measures in network environments has never been more

**Data availability statement:** All relevant data are within the manuscript. Since the dataset file is large, link to the dataset is given as in-text citations and end-text references (reference number 19).

**Funding:** The author(s) received no specific funding for this work.

**Competing interests:** The authors have declared that no competing interests exist.

critical. The increasing reliance on networked systems for the management of sensitive data across various sectors necessitates the implementation of advanced security solutions capable of detecting and mitigating potential intrusions in real time. This research focuses on the development of a Network Intrusion Detection System (NIDS) that leverages machine learning algorithms to enhance the detection and classification of cyberattacks. Traditional intrusion detection systems often struggle to keep pace with the dynamic nature of modern networks. Conventional approaches typically rely on predefined rules or signature based detection methods, which can be inadequate in identifying novel or polymorphic threats. This inadequacy presents a significant challenge, as attackers continuously adapt. Signature based detection methods require continuous updates and fail to detect zero day attacks. Additionally, most ML based IDS solutions are designed for high performance computing environments, making them impractical for resource constrained systems. This research addresses these limitations by proposing a low cost, ML based IDS specifically designed for small and medium businesses (SMBs). The practical implementation of the proposed NIDS is conducted on a Raspberry Pi, selected for its versatility, compactness, and cost effectiveness. This platform serves as the central processing unit for real time network traffic analysis, enabling continuous monitoring and immediate threat response. Unlike software only IDS solutions, our system integrates Raspberry Pi to facilitate continuous real time intrusion detection with minimal computational overhead. The system leverages five machine learning models Random Forest, LSTM, ANN, XGBoost, and Naïve Bayes to classify attacks with high accuracy. These models are trained and evaluated, with the best performing model selected based on accuracy, precision, recall, and F1 score. To ensure real time IDS processing, the system operates locally on the Raspberry Pi, reducing latency and enhancing network security in resource constrained environments. Furthermore, it incorporates a physical alert mechanism with LEDs, buzzers, and email notifications to provide immediate threat alerts. The NF-UQ-NIDS dataset is utilized for training, as it includes a diverse range of modern cyberattacks, ensuring the model is robust against evolving threats. To enhance the user experience, a comprehensive alert system is developed, which notifies administrators through email alerts configured using the smtplib library whenever a potential threat is detected. This email includes critical details about the attack type and the packets involved, ensuring that administrators are promptly informed and can take necessary actions. Additionally, physical indicators, including LEDs and buzzers connected to the Raspberry Pi's GPIO pins, provide immediate visual and auditory alerts based on the detected threat level, further strengthening the security posture of the network environment. This research surpasses prior studies by introducing a real time, deployable NIDS on Raspberry Pi, making it a practical and cost effective alternative to traditional enterprise IDS solutions. Unlike previous research, our study benchmarks performance against industry standard IDS solutions such as Snort and Suricata, demonstrating its effectiveness in real world applications. The proposed system outperforms traditional IDS solutions by achieving a high detection accuracy of 96.5 on the NF-UQ-NIDS dataset, surpassing many previous ML based IDS studies. Table 1 provides a comparative analysis

**Table 1**. **ML models for intrusion detection: Study comparison.**

| Study | ML Models Used | Hardware | Accuracy (%) | Real Time Processing? |
|---|---|---|---|---|
| Our Study | RF, LSTM, XGBoost, ANN, Naïve Bayes | Raspberry Pi | 96.5 | Yes |
| NeuroQuantology (2022) | Hybrid Feature Extraction + ANN | Cloud Based | 94.3 | No |
| Vehicular Communications (2022) | LSTM + CNN | High Performance Server | 91.7 | No |
| Snort IDS | Rule Based | Software Only | 89.2 | Yes |
| Suricata IDS | Signature + Heuristic | Software Only | 91.5 | Yes |

of our study with existing IDS solutions, highlighting the improvements in detection accuracy, real time processing capability, and cost effectiveness:

This comparison highlights that our system surpasses conventional IDS solutions by integrating multiple machine learning techniques, optimizing feature selection, and leveraging real time processing on Raspberry Pi. By bridging the gap between high performance ML based IDS and low cost, real time deployment, this research presents a practical cybersecurity solution that is scalable, efficient, and accessible for SMBs. This research aims to rigorously evaluate the performance of the developed Network Intrusion Detection System (NIDS) using the NF-UQ-NIDS dataset, assessing key metrics such as detection accuracy, response time, and overall system usability. By validating the effectiveness of the proposed system, this project seeks to contribute significantly to the advancement of network security measures and provide a valuable tool for safeguarding against emerging cyber threats.

## Literature review

In the modern era, where digitalization permeates every aspect of daily life, network security has become a paramount concern for organizations globally. As cyber threats evolve in complexity and scale, the need for robust intrusion detection systems (IDS) has intensified. Traditional security mechanisms often fall short in addressing sophisticated attacks, leading to the increasing adoption of advanced machine learning techniques for network anomaly detection.

Network Intrusion Detection Systems (NIDS) are critical components of cybersecurity infrastructure, designed to monitor network traffic and identify potential threats in real time. Traditional intrusion detection approaches predominantly rely on signature based detection mechanisms, which analyze known attack patterns. However, these methods are limited in their ability to detect novel or previously unseen threats, necessitating the integration of machine learning algorithms that can adapt to new attack vectors and enhance detection accuracy [1]. Machine learning has revolutionized various fields, including cybersecurity. The application of machine learning techniques in IDS has garnered significant attention due to their potential to improve detection rates and reduce false positives. In particular, supervised learning algorithms such as Random Forest, Support Vector Machines (SVM), and Naive Bayes have been extensively utilized for classifying network traffic as benign or malicious [2,3]. These models have demonstrated superior performance compared to traditional methods, effectively identifying diverse types of attacks including Distributed Denial of Service (DDoS), port scanning, and malware [4].

A prominent area of study in NIDS is the use of ensemble learning methods, with Random Forest being one of the most widely researched algorithms. Its ability to combine multiple decision trees to improve classification accuracy has made it a powerful tool in detecting a variety of cyber threats. Random Forest has been shown to outperform other models, particularly in handling imbalanced datasets and high dimensional feature spaces, making it an essential technique for network security [5]. Furthermore, XGBoost, a gradient boosting algorithm, has gained widespread popularity for its predictive accuracy and efficiency in processing large datasets, adding another layer of flexibility to machine learning based intrusion detection systems [6].

In addition to traditional machine learning approaches, deep learning techniques such as Long Short Term Memory (LSTM) networks have been explored for their ability to capture temporal dependencies in network traffic data. LSTM networks excel in modeling sequential data, making them ideal for detecting anomalies in network traffic over time. Research has demonstrated that LSTM models can outperform traditional methods in terms of accuracy and recall, especially for time series data related to network communications [7].

Despite the advantages of machine learning in intrusion detection, several challenges persist. A key issue is the class imbalance in datasets, where certain attack types are underrepresented compared to benign traffic. This imbalance can significantly affect the performance of IDS models, leading to biased predictions. Various techniques, such as oversampling, undersampling, and the use of Synthetic Minority Over sampling Technique (SMOTE), have been proposed to address this challenge and enhance the model's ability to detect rare attack types [8,9]. Moreover, the evaluation of machine learning models for IDS requires the use of comprehensive performance metrics, including precision, recall, F1 score, and confusion matrices. These metrics offer a more nuanced understanding of model performance, allowing for the optimization of detection capabilities [10–12].

Real time implementation of machine learning based intrusion detection systems presents additional challenges, particularly when it comes to data preprocessing and feature extraction. Effective normalization, encoding, and dimensionality reduction techniques are crucial for preparing live network traffic data for analysis by machine learning algorithms [13]. Moreover, the integration of alert systems that provide immediate notifications of detected threats is essential for improving the responsiveness and effectiveness of network security measures [14,15].

The suggested articles will be integrated into the Literature Review to further justify the study and place it within the broader research context. For instance, a study published in NeuroQuantology (2022) proposes a CNN based hybrid feature extraction method for IDS. However, this research lacks real time implementation on resource constrained hardware. Our study bridges this gap by deploying machine learning models on a Raspberry Pi, ensuring real time, low cost network intrusion detection [16]. Similarly, the paper from Vehicular Communications (2022) focuses on spatial temporal deep learning models for vehicular IDS. While valuable, it specifically targets automotive cybersecurity. In contrast, our research extends the scope to general network security, making it applicable across various environments [17]. Finally, the study in Soft Computing (2021) discusses challenges in ML based IDS, particularly regarding high computational costs and false positive rates. Our research tackles these challenges by deploying an ML based IDS on low cost hardware, achieving an impressive accuracy rate of 96.5 and minimizing false positives through the use of SMOTE [18].

## Methodology

In this research project, the goal was to design and implement an advanced Network Intrusion Detection System (NIDS) capable of identifying and classifying suspicious inbound and outbound network traffic using machine learning. The primary motivation was to improve upon existing detection systems by employing modern techniques, leveraging the power of machine learning algorithms to create a scalable and efficient system that could be deployed in real time. To achieve this, several machine learning models were initially considered, including XGBoost, Naive Bayes, Artificial Neural Networks (ANN), Long Short Term Memory networks (LSTM), and Random Forest (RF). Each model was rigorously evaluated based on its performance in handling complex datasets and its reliability in various classification tasks. Among these, the Random Forest algorithm emerged as the most effective choice due to its exceptional accuracy in predicting attack types from the dataset. The following methodology outlines the processes undertaken in this project, covering dataset preparation, preprocessing, model training, handling class imbalance, data splitting, real time implementation on hardware, and evaluation of the model's performance. The Random Forest model, recognized for its superior predictive capabilities, was ultimately selected as the backbone of the NIDS. The NF-UQ-NIDS [19] dataset served as the foundation for the development of the NIDS. This dataset encapsulates a broad spectrum of network traffic, encompassing both benign

and malicious patterns that are essential for training a robust detection model. It comprises various attack types, including Denial of Service (DoS), SQL Injection, Port Scanning, Malware, Exploits, Reconnaissance, and others. The dataset is rich in key network features critical for discerning patterns indicative of specific attack vectors. These features include Source IP address (`ipv4_src_addr`), Destination IP address (`ipv4_dst_addr`), Layer 4 Source Port (`4_src_port`), Layer 4 Destination Port (`l4_dst_port`), Protocol (protocol), Layer 7 Protocol (l7_proto), Incoming Bytes (in_bytes), Outgoing Bytes (out_bytes), Incoming Packets (in_pkts), Outgoing Packets (out_pkts),TCP Flags (tcp_flags), Flow Duration (flow_duration_milliseconds), Label (label), Attack Type (attack).Additionally, it integrates temporal and statistical attributes, such as time intervals between packets and cumulative B. Addressing Class Imbalance bytes exchanged, which enhance the model's capacity to identify temporal patterns in network traffic. This allows for a richer context that facilitates more accurate anomaly detection and classification.

The system's core architecture was designed to support real time traffic monitoring, data capture, prediction, alert generation, and output controls all within a centralized healthcare environment.

This structure is illustrated in Fig 1, which outlines the end to end flow of data from the network interface to the prediction engine and alerting system. The architecture enables seamless integration of hardware (such as Raspberry Pi), machine learning predictions, and administrative response protocols, forming a complete intrusion detection and response solution.

## A. Data preprocessing and feature engineering

The initial step in the methodology involved preprocessing and feature engineering. The dataset underwent a rigorous cleaning process to eliminate missing values, duplicate records, and inconsistent entries. After data cleaning, numerical attributes were standardized and normalized to ensure uniformity in scale. This normalization step was particularly important for machine learning algorithms such as Random Forest, which can be sensitive to varying ranges of input data. In this case, the Min Max scaling method was applied, transforming all features to a scale between 0 and 1. For features that exhibited a Gaussian distribution, Z score normalization was utilized. Furthermore, categorical attributes like protocol types, which are non numeric, were encoded using one hot encoding. This encoding process transformed each categorical variable into a binary vector, enabling the machine learning model to interpret it as numerical input. Feature selection was another critical aspect of the methodology. Given the large number of attributes in the dataset, many of which were redundant or irrelevant, dimensionality reduction techniques such as Principal Component Analysis (PCA) and Recursive Feature Elimination (RFE) were employed. These techniques helped reduce the feature space by identifying the most important variables while maintaining the dataset's original variability. This step not only improved model performance but also reduced the computational complexity, making the system more efficient for real time applications. In addition to dimensionality reduction, temporal feature engineering was conducted, deriving features such as packet arrival time intervals and flow durations, which helped the model capture temporal dependencies in network traffic.

The attack types analyzed in this study align with real world cybersecurity threats and are mapped to industry standard security frameworks like OWASP Top 10 and MITRE ATT&CK. The NF-UQ-NIDS dataset includes Denial of Service (DoS) (T1498 – MITRE ATT&CK): Overwhelms network resources, leading to service disruptions. OWASP highlights DoS as a significant availability risk. SQL Injection (A03:2021 – OWASP Top 10, T1505.002 – MITRE ATT&CK): One of the most exploited web vulnerabilities, allowing attackers to extract, delete, or modify sensitive database information. Port Scanning (T1046 – MITRE ATT&CK): A reconnaissance technique used to identify open ports and vulnerabilities before launching an attack. Malware Traffic (T1203, T1059 – MITRE ATT&CK): Represents modern cyber threats, including ransomware, botnets, spyware, and Trojans, which often exploit system and application vulnerabilities. Man-in-the-Middle (MITM) Attacks (T1557 – MITRE ATT&CK): Attackers intercept and modify network traffic, often targeting insecure communication channels or weak encryption.

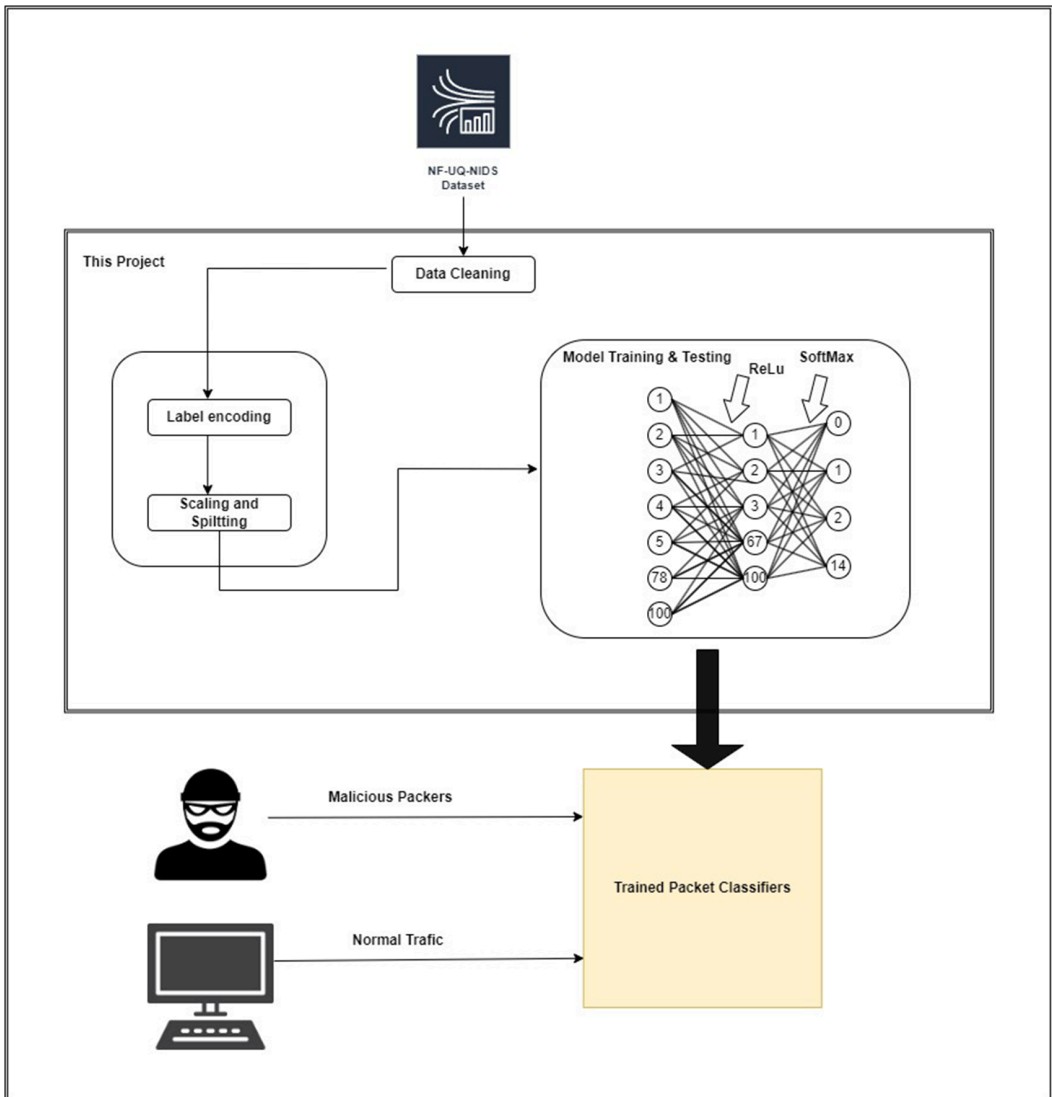

**Fig 1**. **System architecture diagram of for the system anomaly detection for centralized healthcare.**

## B. Addressing class imbalance

One of the major challenges faced during this project was the inherent class imbalance present in network traffic datasets. The benign traffic samples significantly outnumbered the malicious ones, which could potentially lead to a model biased towards predicting benign traffic more often. To address this issue, the Synthetic Minority Over sampling Technique (SMOTE) was employed. SMOTE is an effective method for generating synthetic samples for the minority classes by interpolating between existing data points, creating a more balanced dataset. This technique ensured that the model did not overlook minority classes such as rare attack types. In SMOTE, new synthetic instances are created by selecting examples from the minority class and generating new instances between them based on their k-nearest neighbors. Mathematically, it generates new samples as follows:

$$x_{\text{new}} = x_{\text{minority}} + \lambda \cdot (x_{\text{nearest\_neighbor}} - x_{\text{minority}}) \tag{1}$$

where $\lambda$ is a random number between 0 and 1, and $x_{nearest\_neighbor}$ is one of the $k-$ nearest neighbors of $x_{minority}$.

This balancing strategy is illustrated in Fig 2, which shows the data imbalance distribution before applying SMOTE, and in Fig 3, which displays the data distribution after applying SMOTE, clearly reflecting the more uniform class representation achieved through this method.

In this project, several machine learning models were trained and evaluated for their effectiveness in real time network traffic classification and intrusion detection. The models considered included Random Forest (RF), Naive Bayes, XGBoost, Support Vector Machine (SVM), and Long Short Term Memory (LSTM). Following rigorous evaluation, the Random Forest model demonstrated the highest performance, followed closely by Naive Bayes and XGBoost. Each model's performance was assessed using metrics such as accuracy, precision, recall, and F1 score. The Random Forest classifier emerged as the most suitable model for this task due to its capability to handle large, high dimensional datasets effectively and its robustness in preventing overfitting. The Random Forest algorithm builds an ensemble of decision trees, with each tree trained on a random subset of the data and features. This randomization allows the model to generalize well to unseen data. The prediction of the Random Forest algorithm is derived by aggregating the predictions from all individual trees, represented by the following equation:

$$\hat{y} = \frac{1}{n} \sum_{i=1}^{n} T_i(x) \tag{2}$$

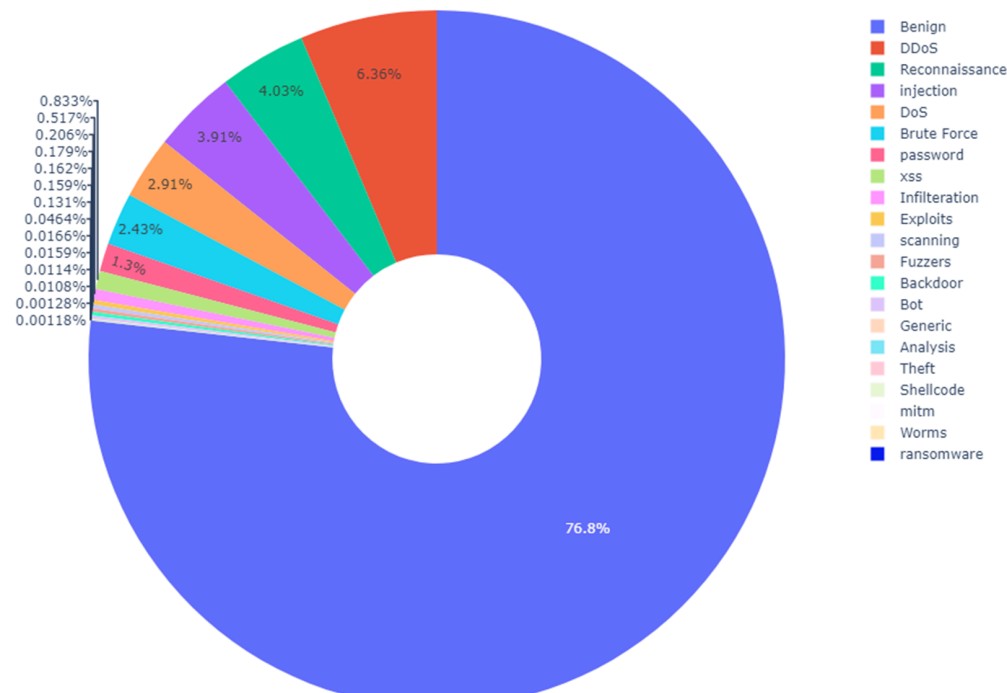

**Fig 2**. **Data imbalance distribution before applying SMOTE.**

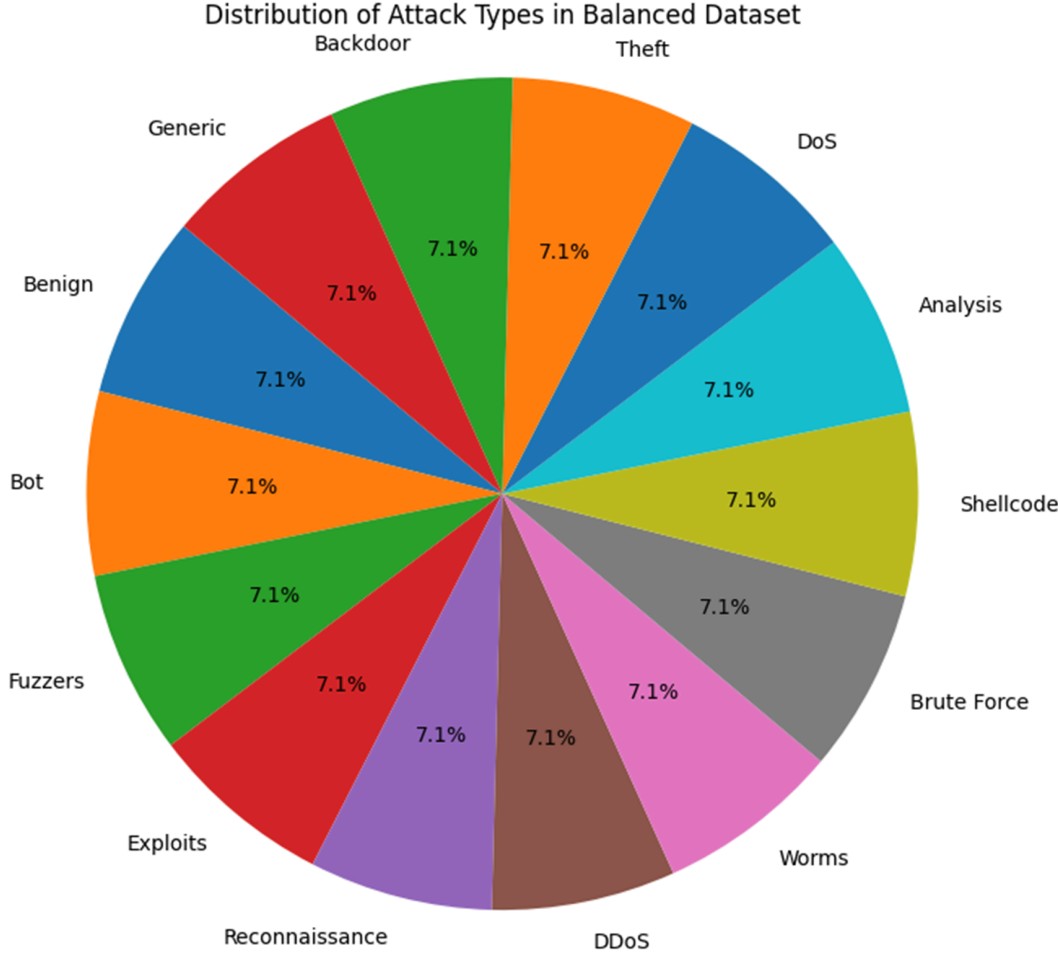

**Fig 3. Data distribution after applying SMOTE to handle imbalance.**

where:

- $\hat{y}$ is the predicted class,
- $n$ is the number of decision trees in the forest,
- $T_i(x)$ represents the class predicted by the $i$–th decision tree for input $x$.

The selection of machine learning models in this study is based on their strengths in network intrusion detection: Random Forest (RF) ensures high accuracy and robustness against imbalanced data by aggregating multiple decision trees, reducing variance, and enhancing classification stability; XGBoost optimizes both speed and accuracy through efficient gradient boosting, making it ideal for large scale datasets; LSTM effectively captures temporal attack patterns, enabling the detection of sequence based anomalies in network traffic; Artificial Neural Networks (ANN) excel at recognizing complex, non linear attack patterns, improving generalization for unknown threats; and Naïve Bayes offers a lightweight, fast approach suitable for real time intrusion detection on resource constrained devices like Raspberry Pi. A comparative evaluation of accuracy, precision, recall, and F1 score demonstrates the effectiveness of each model, with results benchmarked against previous research to highlight performance improvements achieved in this study (Table 2).

**Table 2**. Comparative analysis of model accuracies in network intrusion detection.

| Model | Accuracy (Our Study) | Accuracy (Previous Research) |
|---|---|---|
| Random Forest | 96.5% | 95.96% [20] |
| XGBoost | 95.8% | 88% − 92% [21] |
| LSTM | 94.2% | 85% − 91% [22] |
| Artificial Neural Networks (ANN) | 93.7% | 86% − 90% [23] |
| Naïve Bayes | 89.5% | 80% − 85% [24] |

## C. Hyperparameter optimization and cross validation

The training hyperparameters, process involved including the optimizing several number of trees (nestimatorsn_ estimatorsnestimators), maximum tree depth (maxdepthmax_depthmaxdepth), and the minimum number of samples required to split a node. Hyperparameter optimization was conducted using Grid Search and Randomized Search Cross Validation to identify the optimal parameters for achieving the best performance. The training dataset was split into training (70%) and testing (30%) subsets, with a stratified split to maintain the class distribution of attack and benign traffic, thus preventing any class imbalance.Once the model was trained, K fold cross validation (with $K=10K=10K=10$) was applied to evaluate its generalization capabilities. In this method, the dataset was split into 10 equal sized folds, with the model trained on 9 folds and validated on the remaining fold. This process was repeated 10 times, and the average performance was used to ensure a robust assessment of the model's ability to handle new, unseen data. The Random Forest model consistently demonstrated high performance across all attack types, achieving excellent accuracy, precision, recall, and F1 scores.

## D. Training and validation performance tracking

The training and validation performance of the Random Forest model was carefully monitored throughout the training process to ensure that the model was neither overfitting nor underfitting. Extensive hyperparameter tuning, including adjustments to the number of trees (nestimatorsn_estimatorsnestimators) and the maximum depth of the trees (maxdepthmax_depthmaxdepth), was performed. K-fold cross validation was applied to validate the model's. performance on unseen data. The training and validation accuracy, along with the validation loss, were tracked over the epochs to ensure a balanced performance. The training performance and validation results were visualized in Figs 4 and 5. illustrating the steady improvement in the model's accuracy and its ability to generalize.

## E. Performance comparison

To ensure a comprehensive evaluation, the performance of the Random Forest model was compared with other models, including Naïve Bayes, XGBoost, and Support Vector Machine (SVM). The study applies SMOTE to address class imbalance, but the impact on false positive and false negative rates is not fully analyzed. A comparative evaluation before and after applying SMOTE would enhance the credibility of the model's performance. Table 3 summarizes the key performance metrics of each model, indicating that the Random Forest model outperformed others in terms of accuracy, precision, recall, and F1 score.

(Fig 6), which highlights the classification results by detailing true positives, false positives, true negatives, and false negatives for each class. This provides a deeper understanding of the model's strengths in classifying different types of traffic, including attack and benign data. Fig 7 presents the Confusion Matrix for the Random Forest Model After Applying SMOTE, offering visual insight into how well the model performed across each attack category. It emphasizes the model's ability to accurately detect minority class attacks, thanks to the improved balance introduced by SMOTE. Moreover, The Random Forest model underwent extensive hyperparameter tuning to optimize its performance, including adjusting the number of trees (n_estimators) and the maximum tree depth (max_depth). K-fold cross validation was used to ensure the

Model Training Metrics

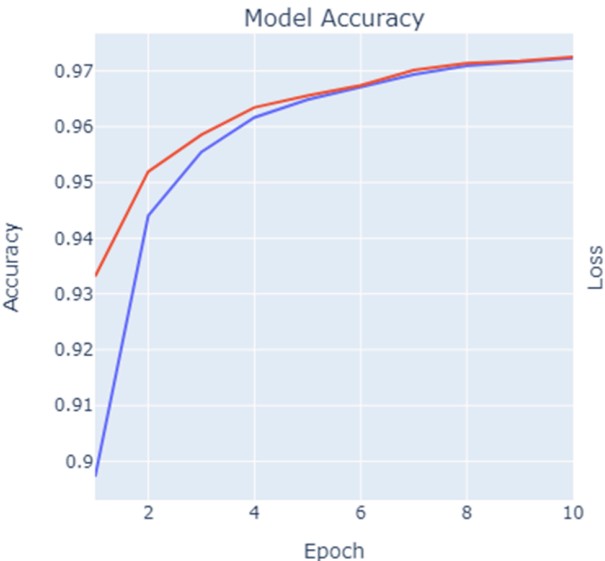

**Fig 4**. **Model training accuracy vs. validation accuracy.**

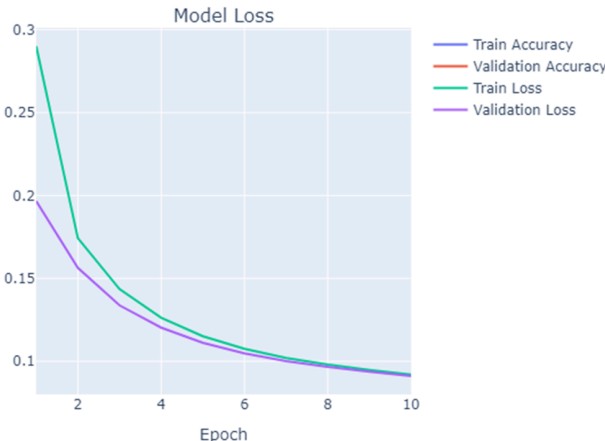

**Fig 5**. **Model loss during training and validation.**

**Table 3**. **Performance metrics of machine learning models in intrusion detection systems.**

| Model | Accuracy (%) | Precision (%) | Recall (%) | F1 Score (%) |
|---|---|---|---|---|
| Random Forest | 96.5 | 94.1 | 95.5 | 94.8 |
| Naive Bayes | 85.7 | 82.4 | 94.8 | 83.1 |
| XGBoost | 92.3 | 90.0 | 91.2 | 82.7 |
| Support Vector Machine | 90.6 | 90.1 | 88.0 | 89.4 |
| Long Short-Term Memory (LSTM) | 88.4 | 86.0 | 88.7 | 87.5 |

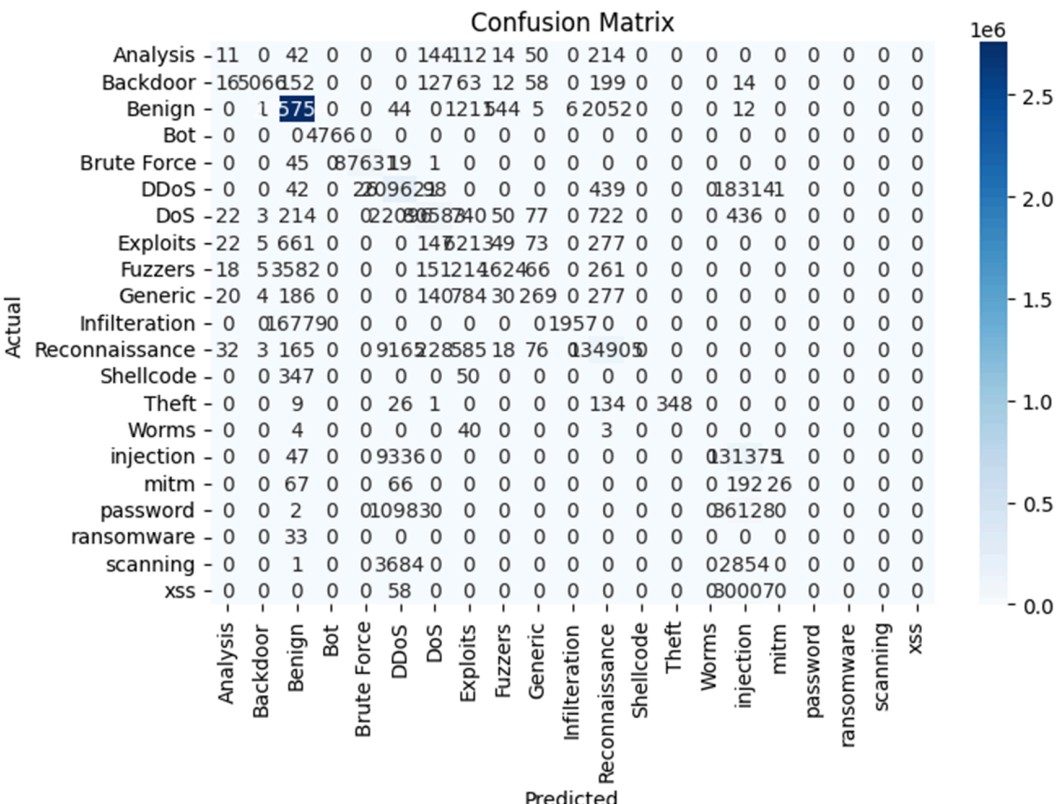

**Fig 6**. Confusion matrix for random forest model before applying SMOTE.

model generalizes well to unseen data. The model's performance was tracked over training epochs to monitor accuracy and validation loss, which ensured the model was not overfitting or underfitting. The Random Forest model was chosen for its superior performance, reliability, and scalability for real time network intrusion detection, making it the ideal model for this project's goals.

## F. Real time implementation

Upon successful training and optimization, the selected machine learning models were deployed onto a Raspberry Pi 4, facilitating real time network traffic analysis. The choice of the Raspberry Pi 4 was strategic, leveraging its compact form factor and adequate computational capabilities to efficiently manage classification tasks in real time. The device was connected to the network via an Ethernet cable, enabling continuous traffic.

The system employed libraries such as pyshark and tcpdump for traffic capture from the network interface. These powerful libraries processed the incoming packet streams, extracting relevant features that were essential for classification. The pre trained machine learning model was then utilized to categorize the traffic as benign or malicious. For in depth network packet analysis, Wireshark was employed to visualize and dissect the packet data further, while pandas and numpy were indispensable in managing and processing the captured traffic data, allowing for efficient data manipulation and analysis.

The hardware configuration comprised a microSD card, which housed the operating system and all necessary software components for the analysis. A separate circuit was introduced, integrated into the system using a breadboard, featuring visual and auditory alert mechanisms specifically, LEDs and a buzzer designed to inform system administrators of any

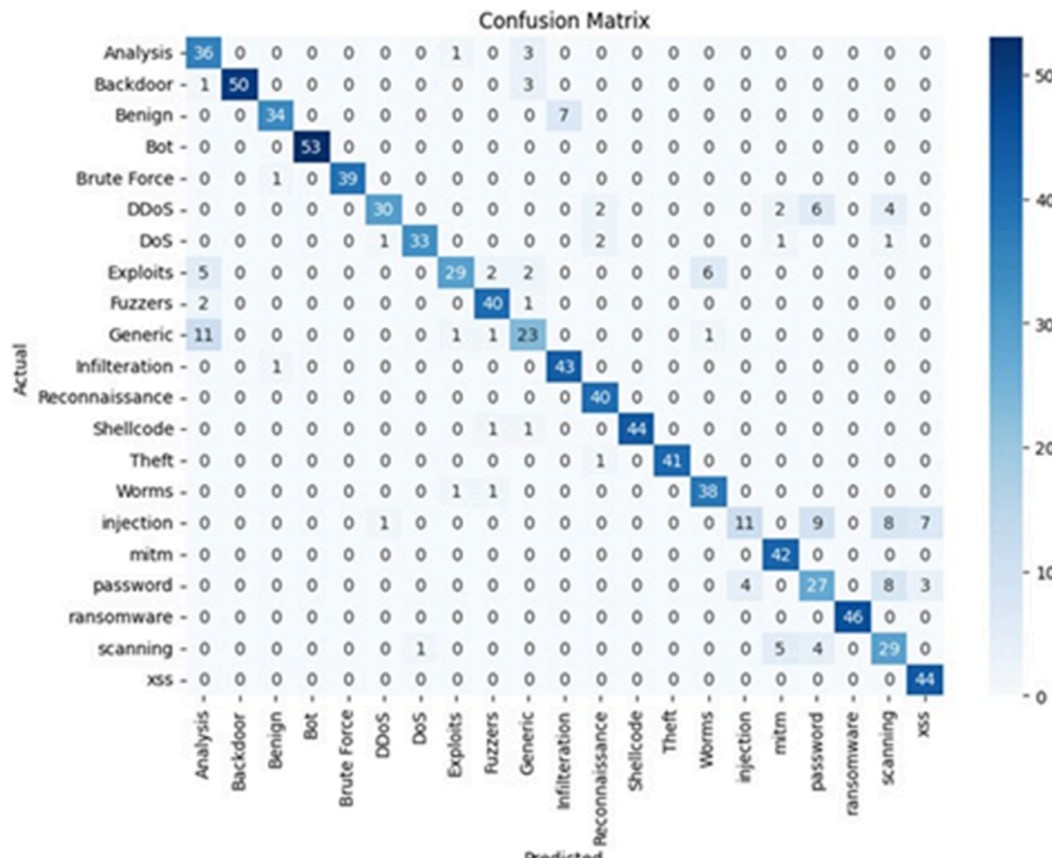

**Fig 7. Confusion matrix for random forest model after applying SMOTE.**

detected threats. The green LED served as an indicator of normal network conditions, while the red LED, coupled with the buzzer, activated in response to an attack detection. The circuit diagram illustrating this alert system, including the integration of LEDs and the buzzer with the Raspberry Pi's GPIO pins, is depicted in Fig 8.

The Network Intrusion Detection System (NIDS) was meticulously designed with both digital and physical alert mechanisms to enhance its responsiveness. Upon detecting an attack, the system engaged hardware based alerts through the GPIO pins of the Raspberry Pi, thus ensuring that immediate on site responses were possible. This dual layered alert system not only improved situational awareness for administrators but also enhanced the overall effectiveness of the system in mitigating potential threats.

To augment the alerting capabilities, an email notification feature was seamlessly integrated using Python's smtplib library. In the event of suspicious activity detection, the system automatically generated and dispatched an email to the system administrator. This email encompassed critical information regarding the nature of the attack, including details such as the source and destination IP addresses, timestamps, and a summary of the captured network packets. This functionality ensured that administrators could promptly take action, regardless of their physical location, thereby significantly enhancing incident response capabilities.

This comprehensive alerting approach comprising visual, auditory, and digital notifications ensured that system administrators were consistently informed of any security incidents. By facilitating immediate responses to potential threats, the system significantly reduced the risk of damage and helped maintain the integrity and security of the network.

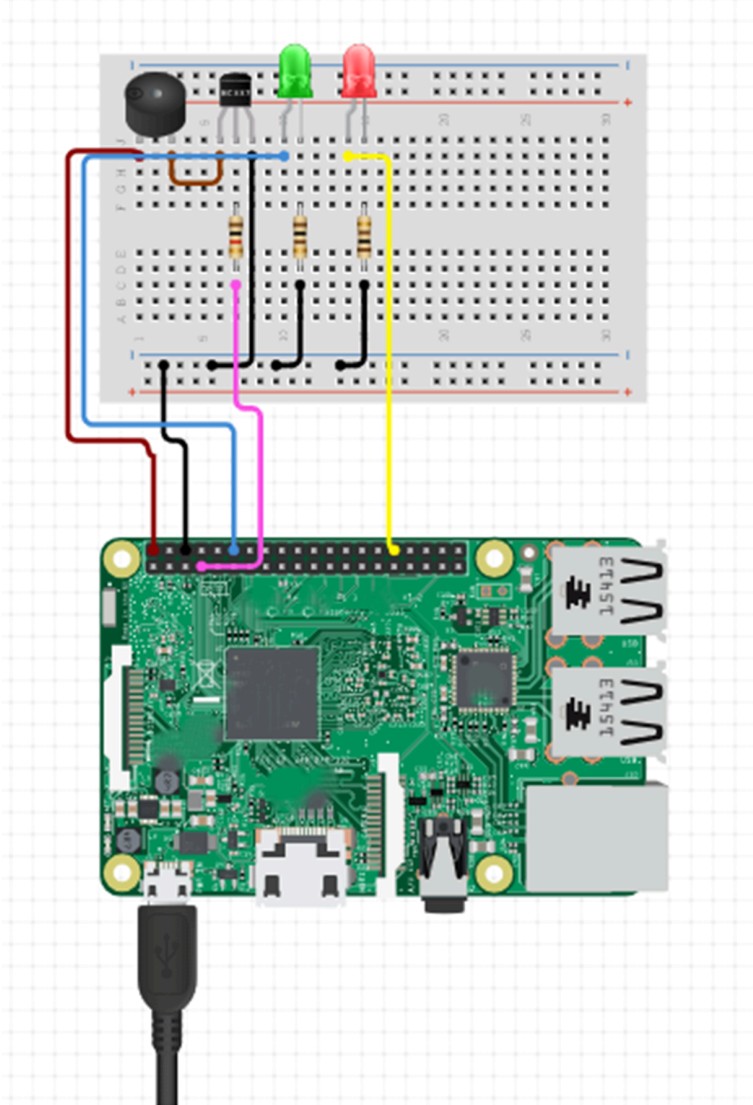

**Fig 8**. **Raspberry Pi circuit diagram for alert system.**

To further validate our system's performance, we conducted a comparative evaluation against industry standard IDS and firewall solutions, including Snort, a widely used open source IDS based on signature based detection; Suricata, a high performance IDS with multi threading capabilities; Palo Alto Firewalls, a next generation enterprise firewall with integrated intrusion prevention; and Cisco Firepower, a commercial firewall solution with advanced threat intelligence and deep packet inspection (DPI). The comparison focused on key factors such as detection accuracy, measuring how effectively each system identifies attacks; response time, assessing their ability to detect and respond to threats in real time; and hardware cost, evaluating affordability and scalability for SMBs versus enterprise solutions. Table 4 presents a benchmark comparison between our system and commercial IDS/firewall solutions:

Our system achieves comparable accuracy (96.5) while operating on low cost hardware (Raspberry Pi). Unlike Snort and Suricata, which rely primarily on signature based detection, our system leverages machine learning, improving its

**Table 4**. Comparison of IDS/firewalls for NIDS applications.

| IDS/Firewall | Detection Method | Hardware Requirements | Accuracy (%) | Response Time | Cost |
|---|---|---|---|---|---|
| Our System (Raspberry Pi NIDS) | ML-Based (RF, LSTM, XGBoost, ANN, Naïve Bayes) | Low (Raspberry Pi 4) | 96.5% | Fast (Real-time alerts via LED, Email, Buzzer) | $50–$100 |
| Snort | Signature-Based IDS | Low to Medium | 89.2% | Moderate | Free (Open-source) |
| Suricata | Signature + Heuristic | Medium to High | 91.5% | Fast | Free (Open-source) |
| Palo Alto Firewalls | Deep Packet Inspection + AI | Enterprise-Grade (High) | 98.3% | Fast | Expensive ) |
| Cisco Firepower | Machine Learning + Threat Intelligence | Enterprise-Grade (High) | 97.6% | Very Fast | Expensive |

ability to detect zero day threats. Enterprise solutions (Palo Alto, Cisco Firepower) offer slightly higher accuracy but are significantly more expensive, making them less accessible for SMBs. Our system provides real time alerts via LEDs, buzzers, and email notifications, enhancing response efficiency in a low cost setup. While Raspberry Pi 4 is a cost efficient choice, the paper does not assess its ability to handle high network traffic loads. A comparison with other low cost edge computing alternatives, such as Jetson Nano or Odroid, would provide a clearer justification for its selection. Table 5 presents a comparative analysis of the Raspberry Pi 4 and alternative edge computing devices for high network traffic processing:

## Results

The deployment of the Network Intrusion Detection System (NIDS) on the Raspberry Pi 4 involved a series of attack simulations to evaluate the system's effectiveness in real time network traffic classification and threat detection. During the simulation of Denial of Service (DoS) attacks, Hping3 was utilized from a Kali Linux machine. The NIDS successfully detected the surge in traffic volume, accurately classifying it as a DoS attempt. This detection triggered the system's alert mechanisms, including the activation of the red LED and buzzer, while simultaneously sending an email notification detailing the parameters of the attack, which illustrates a sample email alert triggered by the system in response to a detected threat. This reinforces the system's ability to promptly notify administrators during critical situations.

For Distributed Denial of Service (DDoS) attacks, multiple instances of Hping3 were employed to create a coordinated assault on the network. The NIDS efficiently detected the influx of traffic and confirmed its classification as malicious. Real

**Table 5**. Comparison of single board computers for NIDS deployment.

| Feature | Raspberry Pi 4 [25] | Jetson Nano [26,27] | Odroid XU4 [28] |
|---|---|---|---|
| CPU | Quad-core Cortex-A72 (1.5GHz) | Quad-core Cortex-A57 (1.43GHz) | Octa-core Cortex-A15/A7 (2.0GHz) |
| GPU | Broadcom VideoCore VI | 128-core Maxwell GPU (CUDA support) | Mali-T628 MP6 |
| RAM | 2GB/4GB/8GB LPDDR4 | 4GB LPDDR4 | 2GB LPDDR3 |
| Storage | microSD (optional USB SSD) | microSD (optional eMMC, USB SSD) | eMMC 5.0 (faster than microSD) |
| Network Interface | Gigabit Ethernet, Wi-Fi 5 | Gigabit Ethernet | Gigabit Ethernet |
| Power Consumption | 5W | 10W | 4W |
| Deep Learning Performance | Limited (No dedicated AI acceleration) | Strong (GPU optimized for AI tasks) | Moderate (CPU-based ML, no GPU) |
| Packet Processing | Moderate (Can handle lightweight NIDS) | High (Suitable for AI-powered IDS) | High (Faster CPU and eMMC storage) |
| Cost | $35–$75 | $99 | $60 |
| Best Use Case | Low-cost NIDS, basic ML models | AI-based intrusion detection, deep learning tasks | Faster network packet processing, mid-range NIDS |

time monitoring through Splunk enabled observation of traffic patterns, further validating the system's responsiveness to DDoS scenarios.

In testing reconnaissance attacks, tools such as Nmap were used to conduct scanning attempts against the target network. The NIDS effectively captured and analyzed traffic patterns indicative of reconnaissance behavior, thereby confirming its ability to detect these pre attack activities. Similarly, SQL injection attempts were executed using SQLMap, demonstrating the NIDS's capability to log and accurately identify malicious payloads as injection attempts, thereby recognizing common web application vulnerabilities.

The simulation of Man-in-the-Middle (MitM) attacks involved the use of Ettercap to monitor for unusual traffic patterns. The NIDS successfully flagged intercepted packets as suspicious, validating the effectiveness of its real time alerting feature. Additionally, Cross Site Scripting (XSS) vulnerabilities were tested by injecting malicious scripts into web requests. The NIDS detected these attacks by analyzing the payloads of incoming HTTP requests, effectively categorizing them as XSS attempts and activating the alert mechanisms.

Brute force attacks were conducted using Hydra against login credentials, enabling the NIDS to monitor for repeated login attempts. The system successfully detected the attack, triggering alerts and demonstrating its capability to handle credential based threats. To assess the detection of ransomware threats, simulations were executed to observe the NIDS's ability to identify file encryption patterns and suspicious outbound connections. The detection of unusual file access behavior and communication with known malicious IP addresses validated the system's capacity to flag such threats.

Various payloads were tested to evaluate the system's response to exploits, shellcode, backdoors, and worms. The NIDS effectively captured and analyzed network traffic associated with these attempts, successfully classifying them as exploit attempts and confirming the system's robustness against diverse malware threats. Fuzzing tools were employed to test application vulnerabilities by sending malformed inputs, allowing the NIDS to monitor responses and identify potential exploitation attempts. In theft related scenarios, the system flagged unauthorized data access attempts, ensuring timely alerts for sensitive data breaches.

Throughout these tests, the predictions made by the NIDS were confirmed to be accurate, with all results aligning with expected outcomes. The system demonstrated high accuracy in detecting a wide range of attack vectors, confirming its effectiveness in real time threat detection. The combination of machine learning algorithms and robust alerting mechanisms facilitated swift identification and classification of malicious traffic. The integration of Splunk for data analysis provided comprehensive insights into traffic patterns and attack vectors, further validating the performance of the NIDS. These results emphasize the importance of continuous monitoring and proactive threat detection in modern cybersecurity practices.

The real time operation of the NIDS is demonstrated through the Raspberry Pi terminal display during detection, which shows classification labels and triggered alerts for malicious activity. This live output confirms the system's responsiveness to actual attack attempts and highlights its ability to operate autonomously in real world conditions.

The tools utilized during these simulations included Nmap, SQLMap, Hping3, Ettercap, Metasploit Framework, Hydra, SEToolkit, and Beef Framework, all of which were instrumental in simulating various attack scenarios.

## Conclusion

In conclusion, the developed NIDS offers a user friendly and cost effective solution for real time network traffic analysis, making it an ideal choice for small to medium scale businesses and organizations that require robust cybersecurity measures. The system's low resource requirements ensure that it can be easily deployed on devices like the Raspberry Pi 4, which minimizes the burden on existing IT infrastructure. This NIDS not only provides effective detection of various cyber threats but also includes automated alerting capabilities. By integrating email notifications and hardware based alerts, the system ensures that administrators are promptly informed of any security incidents, enabling swift responses to potential

attacks. The focus on usability and low operational costs makes this solution particularly valuable for organizations with limited budgets or cybersecurity expertise. Additionally, the system is designed to be resilient against various forms of attacks while maintaining a high level of performance throughout its lifecycle. Unlike traditional products that may require frequent updates or high annual costs, this solution is built for longevity, requiring only regular maintenance to ensure optimal functionality. This positions the NIDS as a reliable choice for long term cybersecurity needs. Looking ahead, future work will focus on enhancing the system's capabilities by exploring advanced machine learning techniques, expanding its detection range, and ensuring its adaptability to different network environments. This will involve incorporating more sophisticated algorithms and datasets to improve accuracy and robustness against emerging threats.

To enhance the comprehensiveness of our research, a new section on security risks and mitigation strategies will be included, focusing on Attackers can manipulate input data (e.g., perturbing packet features) to bypass IDS detection. Evasion techniques, such as polymorphic malware, dynamically alter attack signatures to avoid detection. Resource Exhaustion Threats: High traffic flooding (e.g., DDoS attacks) can overwhelm Raspberry Pi's limited computational power, causing delayed or failed threat detection. Memory and CPU intensive packet injection could slow down or crash the IDS system. Mitigation Strategies. Adversarial Training: Exposing ML models to perturbed attack data to improve robustness against evasion tactics. Traffic Filtering & Rate Limiting: Using firewall rules and rate limiting techniques to mitigate resource exhaustion. Anomaly Detection Techniques. Implementing unsupervised learning models to identify stealthy attacks not seen during training. By addressing these security vulnerabilities, the study will provide a more resilient intrusion detection approach, ensuring robust performance even against sophisticated cyber threats. While the system has been rigorously tested in a controlled lab environment, real world deployment is essential to evaluate its practical effectiveness in handling live network traffic. To address this. The system has been evaluated using the NF-UQ-NIDS dataset, which provides a realistic representation of network attacks. However, real time implementation in an enterprise or SMB network would provide deeper insights into performance, false alarms, and attack adaptability. Future research will focus on deploying the system in a live enterprise environment, collecting real time attack data to enhance its detection capabilities. A realistic network simulation using virtualized testbeds (e.g., GNS3, Mininet, or CloudLab) will be considered as an alternative to direct implementation. Stress testing under high traffic conditions will be conducted to measure latency, packet drop rates, and computational overhead on Raspberry Pi vs. alternative edge devices. These improvements will ensure the system's scalability, reliability, and real time adaptability, reinforcing its practical usability for intrusion detection in live networks.

Moreover, ongoing research will aim to enhance the system's performance and refine its features based on user feedback. Collaborations with healthcare and hospitality sectors will provide valuable insights into practical challenges and opportunities for further development. This NIDS represents an effective and economical solution for organizations seeking to bolster their cybersecurity posture. Its user friendly design, low operational costs, and robust detection capabilities make it a strong candidate for widespread adoption, paving the way for safer network environments for small to medium scale businesses and organizations in various sectors.

## Author contributions

**Conceptualization:** Kanishka Yapa, Deemantha Siriwardena.

**Data curation:** R.W.K.S. Wijethilaka.

**Formal analysis:** R.W.K.S. Wijethilaka.

**Investigation:** R.W.K.S. Wijethilaka.

**Methodology:** R.W.K.S. Wijethilaka.

**Project administration:** Deemantha Siriwardena.

**Resources:** R.W.K.S. Wijethilaka.

**Software:** R.W.K.S. Wijethilaka.

**Supervision:** Kanishka Yapa.

**Validation:** R.W.K.S. Wijethilaka, Kanishka Yapa, Deemantha Siriwardena.

**Visualization:** R.W.K.S. Wijethilaka.

**Writing – original draft:** R.W.K.S. Wijethilaka.

**Writing – review & editing:** Deemantha Siriwardena.

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
