## [Decision Letter · Decision Letter 0]

13 Feb 2025

PONE-D-24-48249Ultra-Low-Cost, Maintenance-Free, High-Precision, Portable Hardware Firewall for Small to Medium Sized Businesses and Organizations Handling Critical Computer System InfrastructuresPLOS ONE

Dear Dr. Siriwardana,

Thank you for submitting your manuscript to PLOS ONE. After careful consideration, we feel that it has merit but does not fully meet PLOS ONE’s publication criteria as it currently stands. Therefore, we invite you to submit a revised version of the manuscript that addresses the points raised during the review process. Major Revision required.

We look forward to receiving your revised manuscript.

Kind regards,

Elochukwu Ukwandu, PhD

Academic Editor

PLOS ONE

Journal Requirements:

4. We note that Figures 1 and 7 in your submission contain copyrighted images. All PLOS content is published under the Creative Commons Attribution License (CC BY 4.0), which means that the manuscript, images, and Supporting Information files will be freely available online, and any third party is permitted to access, download, copy, distribute, and use these materials in any way, even commercially, with proper attribution. For more information, see our copyright guidelines: http://journals.plos.org/plosone/s/licenses-and-copyright.

a. You may seek permission from the original copyright holder of Figures 1 and 7 to publish the content specifically under the CC BY 4.0 license. 

Reviewers' comments:

Reviewer's Responses to Questions

**Comments to the Author**

1. Is the manuscript technically sound, and do the data support the conclusions?

Reviewer #1: Partly

Reviewer #2: Yes

Reviewer #3: Yes

2. Has the statistical analysis been performed appropriately and rigorously?

Reviewer #1: No

Reviewer #2: No

Reviewer #3: I Don't Know

3. Have the authors made all data underlying the findings in their manuscript fully available?

Reviewer #1: No

Reviewer #2: No

Reviewer #3: Yes

4. Is the manuscript presented in an intelligible fashion and written in standard English?

Reviewer #1: Yes

Reviewer #2: Yes

Reviewer #3: Yes

5. Review Comments to the Author

Reviewer #1: The manuscript requires major revisions to enhance its clarity, rigor, and contribution to the field. The title is overly broad and should be revised to reflect the core contributions. The introduction does not clearly articulate the specific research gaps addressed, and the novelty of the proposed system is not sufficiently highlighted, particularly regarding the integration of machine learning techniques (Random Forest, LSTM, ANN, XGBoost, Naive Bayes) and hardware components like Raspberry Pi for real-time analysis. The benefits and suitability of these techniques should be better justified, and a detailed comparison with existing studies is needed to emphasize how this work advances the state-of-the-art. Furthermore, the abstract, introduction, and conclusion must include key results, such as detection accuracy and response time improvements demonstrated on the NF-UQ-NIDS dataset, to underscore the system’s effectiveness. Addressing these issues will significantly strengthen the manuscript and its impact.

Authors can refer to following articles. 1) Neural Network-based Hybrid Feature Extraction Method for Network Intrusion Detection Systems, NeuroQuantology, 20 (8), pp 427-444, 2022 2) A hybrid deep learning based intrusion detection system using spatial-temporal representation of in-vehicle network traffic, Vehicular Communications 35 (2022) 100471 3) Machine learning and deep learning methods for intrusion detection systems: recent developments and challenges, Soft Computing, 25, pages9731–9763 (2021)

Reviewer #2: This work represents a typical course project, there is no much novelty. Using a Raspberry-Pi board is not really using HW (it is still a SW-based solution) .. All types of attacks considered are really old and based on packet headers .. thus the features utilized are naive. Also, the implementation seems doubtful, how did you get the NIC of the Raspberry PI to readout all 'network traffic'?

Most paragraphs are too long .. ref [16] is missing

Reviewer #3: 1- The study tackles a crucial cyber security challenge by introducing a cost-effective, machine-learning-driven network intrusion detection system (NIDS) tailored for small to medium-sized businesses. The methodology is logically organized, offering a step-by-step explanation of dataset preprocessing, feature selection, and model training. The rationale for using Random Forest, XGBoost, LSTM, and Naïve Bayes is well articulated. The system integrates email notifications alongside physical indicators (LEDs and buzzers), enhancing real-time intrusion detection and response, making it highly practical for business environments.

2- The claim that the system is "maintenance-free" could be misleading, as regular updates for machine learning models, security patches, and hardware upkeep are still necessary. A more precise title would better reflect the system’s actual characteristics.

3- The study demonstrates the system’s effectiveness using the NF-UQ-NIDS dataset but lacks a comparative evaluation against existing commercial or open-source firewall solutions. Including a benchmark against industry-standard firewalls (e.g., Snort, Suricata, Palo Alto, Cisco Firepower) would strengthen the paper’s findings.

4- While Raspberry Pi 4 is a cost-efficient choice, the paper does not assess its ability to handle high network traffic loads. A comparison with other low-cost edge computing alternatives, such as Jetson Nano or Odroid, would provide a clearer justification for its selection.

5- The study applies SMOTE to address class imbalance, but the impact on false positive and false negative rates is not fully analyzed. A comparative evaluation before and after applying SMOTE would enhance the credibility of the model’s performance.

6- The system is primarily tested in controlled environments, without implementation in an actual business setting. Conducting real-world deployment or stress testing on an operational network would provide valuable insights into its practical effectiveness and usability.

7- The paper contains small formatting inconsistencies, such as extra spaces (e.g., "Analysis, , Random Forest" in the Keywords section). A thorough proofreading would improve the overall readability and professionalism of the document.

8- While the study highlights high detection accuracy, it does not explore potential vulnerabilities, such as adversarial attacks on machine learning models, resource exhaustion threats, or evasion techniques. Including a discussion on these security risks and potential mitigation strategies would enhance the comprehensiveness of the research.

6. PLOS authors have the option to publish the peer review history of their article (what does this mean?). If published, this will include your full peer review and any attached files.

Reviewer #1: No

Reviewer #2: No

Reviewer #3: No

---

## [Author Response · Author response to Decision Letter 1]

30 Apr 2025

Dear Sir

Manuscript was revised according to valuable feedbacks given by the reviewers. Response to Reviewers letter and a Manuscript with Track Changes are attached herewith for your convenience.

Dear Sir

Manuscript was revised according to valuable feedbacks given by the reviewers. Response to Reviewers letter and a Manuscript with Track Changes are attached herewith for your convenience.

All relevant data are within the manuscript. Since the dataset file is large, link to the dataset is given as in-text citations and end-text references (reference number 19).

---

## [Decision Letter · Decision Letter 1]

12 Aug 2025

A Cost-Effective Machine Learning-Based Network Intrusion Detection System Using Raspberry Pi for Real-Time Analysis

PONE-D-24-48249R1

Dear Dr. Siriwardana,

We’re pleased to inform you that your manuscript has been judged scientifically suitable for publication and will be formally accepted for publication once it meets all outstanding technical requirements.

Kind regards,

Elochukwu Ukwandu, PhD

Academic Editor

PLOS ONE

Additional Editor Comments (optional):

Reviewers' comments:

Reviewer's Responses to Questions

**Comments to the Author**

1. If the authors have adequately addressed your comments raised in a previous round of review and you feel that this manuscript is now acceptable for publication, you may indicate that here to bypass the “Comments to the Author” section, enter your conflict of interest statement in the “Confidential to Editor” section, and submit your "Accept" recommendation.

Reviewer #3: All comments have been addressed

Reviewer #4: (No Response)

2. Is the manuscript technically sound, and do the data support the conclusions?

Reviewer #3: Yes

Reviewer #4: (No Response)

3. Has the statistical analysis been performed appropriately and rigorously?

Reviewer #3: Yes

Reviewer #4: (No Response)

4. Have the authors made all data underlying the findings in their manuscript fully available?

Reviewer #3: Yes

Reviewer #4: (No Response)

5. Is the manuscript presented in an intelligible fashion and written in standard English?

Reviewer #3: Yes

Reviewer #4: (No Response)

6. Review Comments to the Author

Reviewer #3: All comments have been addressed.

But why haven't you reviewed the last three years' articles in the literature review?

Reviewer #4: (No Response)

7. PLOS authors have the option to publish the peer review history of their article (what does this mean?). If published, this will include your full peer review and any attached files.

Reviewer #3: No

Reviewer #4: No

---

## [Editor Report · Acceptance letter]

PONE-D-24-48249R1

PLOS One

Dear Dr. Siriwardena,

I'm pleased to inform you that your manuscript has been deemed suitable for publication in PLOS One. Congratulations! Your manuscript is now being handed over to our production team.

Kind regards,

on behalf of

Dr. Elochukwu Ukwandu

Academic Editor

PLOS One